

# Krylov complexity in a natural basis for the Schrödinger algebra

**Dimitrios Patramanis[1⋆] and Watse Sybesma[2†]**

**1** Faculty of Physics, University of Warsaw, ul. Pasteura 5, 02-093 Warsaw, Poland
**2** Science Institute, University of Iceland, Dunhaga 3, 107 Reykjavik, Iceland

⋆ d.patramanis@uw.edu.pl , † watse@hi.is

## Abstract

We investigate operator growth in quantum systems with two-dimensional Schrödinger group symmetry by studying the Krylov complexity. While feasible for semisimple Lie algebras, cases such as the Schrödinger algebra which is characterized by a semi-direct sum structure are complicated. We propose to compute Krylov complexity for this algebra in a *natural* orthonormal basis, which produces a *penta*diagonal structure of the time evolution operator, contrasting the usual tridiagonal Lanczos algorithm outcome. The resulting complexity behaves as expected. We advocate that this approach can provide insights to other non-semisimple algebras.

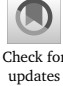

# 1 Introduction

Krylov complexity [1] is a measure introduced in order to quantify how quantum operators change under time evolution. In complex quantum systems one generically expects that, due to the non-trivial dynamics, a typical operator undergoing time evolution will tend to "spread" such that it affects a larger number of the degrees of freedom of the system thereby increasing in complexity. In [2] it was shown that symmetry plays an important role in determining the features of this growth and following similar methods in this work we will be proposing a more general framework that includes symmetries described by non-semisimple Lie groups.

Krylov complexity is conjectured to grow at a maximal rate for chaotic systems, although there are subtleties that need to be taken into account, especially for quantum field theories due to their infinite degrees of freedom in contrast to ordinary quantum mechanical systems, see discussions in [3–10]. When one considers quantum many body systems such as spin chains it is more straightforward to produce evidence that Krylov complexity can for example distinguish between integrable and chaotic dynamics as was argued in [11–13].

However, despite its apparent relevance in the discourse regarding quantum chaos, one might ask why is it worth studying Krylov complexity over all the different available measures of which there has been a profound proliferation in recent years? We believe that there are two main reasons. First, Krylov complexity can be applied to any quantum system making it computationally available, at least in principle, for a plethora of different cases including but not limited to condensed matter and many-body systems [13–17], quantum and conformal field theories [3–5, 18–20], open systems [21–25], topological phases of matter [26, 27] and many other topics related to aspects of the above and not only [28–31]. Second, it is related by its construction to inherent properties and characteristic parameters of the system, namely the Hamiltonian and the Hilbert space that it defines. The reason why this is important is because it removes the arbitrariness which is to a large extent present in other definitions of complexity. Namely, one usually needs to define a set of elementary operations that can be performed on the system and assign a cost to them according to some justifiable albeit arbitrary criteria. For a review on this topic see for example [32, 33]. In the context of Krylov complexity these choices are reduced to the choice of an inner product using which the Lanczos algorithm can map the dynamics of a quantum mechanical system onto a semi-infinite one-dimensional chain with nearest neighbor hopping. Krylov complexity is then simply defined as the average position on the chain as a function of time. In the current note we review some important examples of semisimple algebras for which one can analytically compute Krylov complexity: the case of the Heisenberg-Weyl algebra, corresponding to states expressible in the one-dimensional harmonic oscillator basis, and $SL_2(\mathbb{R})$, which for example represents the two-dimensional conformal algebra.

Subsequently, we will go beyond studying the Krylov complexity of semisimple algebras, which as we will discuss poses certain analytic challenges. We focus on the specific example of the two-dimensional Schrödinger algebra. This algebra is the maximal symmetry algebra for a two-dimensional Schrödinger equation with a quadratic potential or no potential at all and appears as a subalgebra of the three-dimensional conformal algebra $SO(3, 2)$, see e.g. [34]. As the two-dimensional Schrödinger equation is relevant for, e.g., cold atom traps and optical systems [35], having a prediction for Krylov complexity can yield insights in both directions. Moreover, having analytic results for systems with such a symmetry structure can potentially set the groundwork to study more complex systems that are relevant in high energy physics.

However, the usual Krylov approach for the two-dimensional Schrödinger group is problematic from an analytic point of view and only numerical approximations are available [36], making it difficult to work with. In this note we propose an alternative method to probe the Krylov subspace that relies on making use of the semi-direct sum structure of the symme-

try algebra and methods used in the study of coherent states. We find that the semi-direct sum structure presents us with an orthogonal basis that translates to a picture of hopping on a semi-infinite chain where *next-to-nearest* neighbor interactions are allowed, contrasting conventional Krylov complexity approaches. We advocate that generically in the case where symmetry algebras are found to allow for a semi-direct sum structure, it can be more fruitful to use a "natural" orthogonal basis that as a result induces more interactions from the chain point of view, but can allow for direct evaluation.

While this approach in itself is novel, there have been several works already where an orthonormal basis for the Krylov subspace is not obtained through the Lanczos algorithm and as such the Liouvillian (Hamiltonian) is not tridiagonal. These include [18, 37], where (similarly to this work) the symmetry algebra is considered for the full Virasoro group and $SL_n(\mathbb{R})$ respectively. For the former this leads to a block diagonal structure of the Liouvillian and for the latter to a "trivially" non-tridigonal structure along the same lines with $SL_2(\mathbb{R})$ which we examine in more detail in the following sections. Importantly, in both cases, this leads to analytically tractable results, although there does not appear to be a complete understanding of the precise difference between the implementation of this approach as compared to the Lanczos algorithm. Another instance where an explicitly non-tridiagonal form of the Liouvillian is required in order to make progress arises in open quantum systems (see e.g. [23]). The reason is that the Liouvillian for these systems is not a Hermitian operator anymore and as such the Lanczos algorithm is insufficient, so a more general orthogonalization scheme has to be implemented instead. In particular one can use the Arnoldi iteration which puts the Liouvillian in Hessenberg form (triangular matrix with non-zero entries in the first subdiagonal). The common element in all of these considerations is that while it is possible to obtain an orthonormal basis for the Krylov subspace in different ways, the differences between these bases and the associated complexities are unclear. We comment further on this point from the perspective of our results in the discussion section.

The remainder of this note is organized in the following way. In Section 2 we review Krylov complexity and establish notation. In Section 3 we introduce the Schrödinger algebra and present our results. Finally, in section 4 we present a discussion on our obtained results.

## 2  Krylov complexity

In this section we introduce Krylov complexity and establish notation through a series of examples.

### 2.1  Krylov basis

There are many works that include a pedagogical introduction to the notion of the Krylov basis, to which we refer the reader for a detailed derivation [1, 2, 38–40]. The central concept is the expansion of a time evolved operator $\mathcal{O}(t) = e^{iHt}\mathcal{O}e^{-iHt} = e^{i\mathcal{L}t}\mathcal{O}$ in the Krylov subspace $\mathcal{H}_{\mathcal{O}}$ defined as the linear span of the action of the Liouvillian on the initial operator

$$\mathcal{H}_{\mathcal{O}} = \text{span}\{\mathcal{L}^n\mathcal{O}\}_{n=0}^{+\infty}. \tag{1}$$

In other words one seeks to decompose an operator at some arbitrary time as

$$|\mathcal{O}(t)) = \sum_n \phi_n |K_n). \tag{2}$$

Here we adopt the curly bracket notation to denote a state in the operator Hilbert space. Therefore, $|K_n)$ denotes an orthonormal basis for the Krylov subspace and $\phi_n$ the appropriate coefficients satisfying $\sum_n |\phi_n|^2 = 1$. The most common way of obtaining such a basis is

through the Lanczos algorithm which applies the Gram-Schmidt orthogonalization scheme to the elements $|\mathcal{O}_n) = \mathcal{L}^n|\mathcal{O})$ [41]. This has the effect of tridiagonalizing the Liouvillian such that

$$\mathcal{L}|\mathcal{O}_n) = b_{n+1}|\mathcal{O}_{n+1}) + b_n|\mathcal{O}_{n-1}). \tag{3}$$

This tridiagonal structure manifests explicitly when the elements of the Liouvillian are represented in matrix form.

$$\mathcal{L} = \begin{pmatrix} 0 & b_1 & 0 & \cdots & 0 \\ b_1 & 0 & b_2 & \ddots & \vdots \\ 0 & b_2 & 0 & \ddots & 0 \\ \vdots & \ddots & \ddots & \ddots & b_n \\ 0 & \cdots & 0 & b_n & 0 \end{pmatrix}. \tag{4}$$

Note that the diagonal elements are zero due to the properties of the operator inner product, which we assume (as is standard) to involve a trace and given that we are interested in the growth of Hermitian operators. Having obtained the basis one can then study different aspects of the probability distribution provided by $\phi_n$ to probe the growth of the operator in the Krylov subspace and assess its resulting complexity. In particular the average of that distribution $\sum_n n|\phi_n|^2$ is dubbed Krylov complexity and quantifies, albeit in a somewhat crude manner, the growth of the operator. A natural question that arises is why should one consider the average as a measure of complexity rather than any other moment of the distribution. There are some formal arguments to be made in favour of that choice as is done for example in [42], but here we will take a more heuristic approach that will prove useful in our subsequent considerations. The so-called "Krylov chain" picture was pointed out in the seminal work of [1] and it has since been explored by several others [13, 14, 43]. This picture arises from the realization that the dynamics of the Krylov subspace can be mapped to that of a particle hopping on a semi-infinite, one-dimensional chain. This identification is made by the discrete Schrödinger equation that relates the Lanczos coefficients $b_n$ with the coefficients $\phi_n$ as

$$-i\frac{d\phi_n}{dt} = b_{n+1}\phi_{n+1} + b_n\phi_{n-1}. \tag{5}$$

In this picture the $b_n$ play the role of the hopping coefficients between two adjacent sites and $\phi_n$ the amplitude of finding the particle at each site. Krylov complexity can be interpreted as the average position on the chain at a particular time. This is indeed a very useful piece of information, especially in cases where access to other aspects of the probability distribution defined by the $\phi_n$ is limited. However, there is much more refined information that one can extract from said probability distribution if it can be obtained analytically. For instance one can study the Krylov variance [18], entropy [39, 44, 45], logarithmic negativity and capacity of entanglement [43].

Our main motivation is to study the Krylov subspace as it arises from the symmetry of the quantum system of interest in the spirit of [2]. Assuming that this symmetry is described by a Lie group and that the system is closed, then the Liouvillian can be written as a linear combination of the algebra generators. For three-dimensional Lie algebras that admit a representation in terms of some generalized ladder operators one can write the Liouvillian in the following form

$$\mathcal{L} = \xi L_+ - \bar{\xi} L_-. \tag{6}$$

The action of such a Liouvillian on a general Fock state $|n\rangle$ is then by construction identical to how the Liouvillian acts on the Krylov basis and by an appropriate choice of $\xi$ one can identify the two in a precise mathematical manner. This enables us to identify the operator $e^{i\mathcal{L}t}$ as a

group element, thus leading to an interpretation of the time evolved operator

$$|\mathcal{O}(t)) = e^{i\mathcal{L}t}|\mathcal{O}(0)) = \sum_n \phi_n |n\rangle\,, \tag{7}$$

as a coherent state. The latter have been studied extensively [46] and are very well known for a number of different symmetry groups. This is particularly useful given that one can readily obtain the coefficients $\phi_n$ from which the Krylov complexity can be extracted, but it additionally endows these results with a certain degree of universality. More specifically, this construction implies that any system with the same type of symmetry will exhibit the same behaviour in terms of its Krylov complexity. As mentioned in the introduction our aim here is to study a group with more general structure for which the Liouvillian is not automatically tridiagonal. Our vehicle in that endeavour is the Schrödinger group which in 1+1 dimensions is the semi-direct product of the Heisenberg-Weyl and $SL_2(\mathbb{R})$ groups. For that reason, we review the methods and results for these two groups in the following subsections.

## 2.2 Heisenberg-Weyl algebra

The Heisenberg-Weyl algebra ($HW$) is defined by (we borrow the notation employed in [2])

$$[a, a^\dagger] = 1\,, \qquad [\hat{n}, a^\dagger] = a^\dagger\,, \qquad [\hat{n}, a] = -a\,, \tag{8}$$

with all other commutators vanishing, $\hat{n} = a^\dagger a$ and the usual creation and annihilation operators $a^\dagger$ and $a$. Defining a vacuum state $|0\rangle$ via $a|0\rangle = 0$ we can consider the following orthonormal basis for the corresponding Hilbert space:

$$|n\rangle = \frac{1}{\sqrt{n!}}(a^\dagger)^n|0\rangle\,, \tag{9}$$

such that

$$a^\dagger|n\rangle = \sqrt{n+1}|n+1\rangle\,, \qquad a|n\rangle = \sqrt{n}|n-1\rangle\,. \tag{10}$$

To compute the associated Krylov complexity it was shown [2] that we can simply make the following identifications

$$\mathcal{L} = \alpha(a^\dagger + a)\,, \qquad |\mathcal{O}_n) = |n\rangle\,. \tag{11}$$

This allows us the write the Heisenberg operator state in Krylov space as

$$|\mathcal{O}(t)) = e^{i\alpha t(a^\dagger + a)}|0\rangle = \sum_{n=0}^{\infty}(it)^n\phi_n(t)|n\rangle\,, \qquad \phi_n = e^{-\alpha^2 t^2}\frac{\alpha^n t^n}{\sqrt{n!}}\,, \qquad \sum_{n=0}^{\infty}|\phi_n|^2 = 1\,, \tag{12}$$

where at the second equality sign we simply wrote the exponent in its series representation and worked out that $(a^\dagger)^n|0\rangle = \frac{1}{\sqrt{n!}}|n\rangle$. This allows us to compute Krylov complexity $K_{\mathcal{O}}$:

$$K_{\mathcal{O}} = \sum_{n=0}^{\infty} n|\phi_n(t)|^2 = \alpha^2 t^2\,. \tag{13}$$

While the $HW$ group appears elementary it actually arises in cases much more sophisticated than that of the harmonic oscillator with which it is usually associated. For example in [47] the authors discuss how the symmetry of SYK in the triple scaling limit is described by $HW$ and how that leads to the universal Krylov complexity result presented above.

## 2.3 $SL_2(\mathbb{R})$ algebra

We will now consider the $SL_2(\mathbb{R})$ (or the isomorphic $SU(1,1)$) algebra and compute complexity in two different bases: the Heisenberg-Weyl basis and the 'natural' $SL_2(\mathbb{R})$ basis. The algebra is defined as

$$[L_0, L_{\pm 1}] = \mp L_{\pm 1}, \qquad [L_1, L_{-1}] = 2L_0, \tag{14}$$

where we define a vacuum $|h\rangle$ via $L_1|h\rangle = 0$ and $L_0|h\rangle = h|h\rangle$ where positive integer $h$ labels the different states that are allowed. An orthonormal basis is given by

$$|h, n\rangle = \sqrt{\frac{\Gamma(2h)}{n!\Gamma(2h+n)}} L_{-1}^n |h\rangle, \tag{15}$$

such that

$$L_0|h,n\rangle = (h+n)|h,n\rangle, \quad L_{\pm}|h,n\rangle = \sqrt{\left(n + \frac{1\mp 1}{2}\right)\left(2h + n - \frac{1\pm 1}{2}\right)}|h, n \pm 1\rangle. \tag{16}$$

To compute the associated Krylov complexity $K_{\mathcal{O}}$ we can make the following identifications [2]

$$\mathcal{L} = \beta(L_{-1} + L_1), \qquad |\mathcal{O}_n\rangle = |h, n\rangle. \tag{17}$$

It then follows that

$$\phi_n(t) = \sqrt{\frac{\Gamma(2h+n)}{n!\Gamma(2h)}} \frac{\tanh^n(\beta t)}{\cosh^{2h}(\beta t)}, \qquad K_{\mathcal{O}} = 2h\sinh^2(\beta t). \tag{18}$$

What happens when we express the above in the $HW$ basis instead? After some algebra we establish the mapping

$$L_0 = \frac{1}{4}(a^\dagger a + a a^\dagger), \qquad L_{+1} = \frac{1}{2}a^2, \qquad L_{-1} = \frac{1}{2}(a^\dagger)^2, \tag{19}$$

with value of $h = 1/4$, which we treat as a continuation from the usual integer values. As a result the exponentiated Liouvillian takes the form of a so-called squeeze operator which one encounters quite frequently in quantum optics [48]. In that case it is known that [46]

$$|\mathcal{O}(t)\rangle = \sum_{n=0}^{\infty} i^n \frac{\sqrt{(2n)!}}{2^m m!} \frac{\tanh^n(\beta t)}{\sqrt{\cosh(\beta t)}} |2n\rangle, \tag{20}$$

which leads to the same complexity as previously computed, but restricted to $h = 1/4$.

# 3 Two-dimensional Schrödinger algebra

## 3.1 Schrödinger symmetries

The Schrödinger group is the maximal symmetry group corresponding to the free Schrödinger equation [49] and is isomorphic to the maximal symmetry group of the Schrödinger equation with a harmonic potential [50]. In general, from a group perspective the Schrödinger group can be viewed as the extension of the Galilean group. The generators of the Galilean group consist of time ($T$) and spatial translations ($P_i$), Galilean boost ($G_i$) and spatial rotations. The Galilei algebra admits a central extension $M$ of the commutator of the boost generator $G_i$ and the spatial translation generator $P_i$. Adding this extension yields the Bargmann algebra. In order to reach the Schrödinger algebra we take into account a dilatation operator $D$ under which

time scales twice as fast as space and a special conformal symmetry generator $K$. We tailored the names of these generators to the case of the free Schrödinger equation. In two dimensions the only non-vanishing commutators of this algebra are given by (where we dropped the index $i$ as it runs over the spatial indices)

$$[P,G] = -iM\,, \tag{21}$$

$$[D,T] = -2T\,, \qquad [D,K] = 2K\,, \qquad [T,K] = D\,, \tag{22}$$

$$[D,P] = -P\,, \qquad [D,G] = G\,. \tag{23}$$

In (21) we recognize a Heisenberg-Weyl sub-algebra and in (22) we recognize an $SL_2(\mathbb{R})$ sub-algebra. The remaining brackets in (23) are between elements of either sub-algebra. We conclude that the two-dimensional Schrödinger algebra is a semi-direct sum $sl_2(\mathbb{R}) \ltimes hw$.

Using the insight from Section 2 in which we showed that $sl_2(\mathbb{R})$ can be written in terms of $hw$ generators we cast all the algebra elements of the two-dimensional Schrödinger in terms of ladder operators:

$$P = -\frac{i}{\sqrt{2}}(a - a^\dagger), \qquad G = \frac{1}{\sqrt{2}}(a^\dagger + a), \qquad M = aa^\dagger - a^\dagger a\,, \tag{24}$$

$$T = -\frac{1}{4}(a - a^\dagger)^2\,, \qquad K = -\frac{1}{4}(a + a^\dagger)^2\,, \qquad D = \frac{1}{2}(a^2 - (a^\dagger)^2)\,. \tag{25}$$

Therefore, the Liouvillian expressed as a general element of the Lie algebra can be cast in the form

$$\mathcal{L} = \alpha(a^\dagger + a) + \frac{\beta}{2}((a^\dagger)^2 + a^2)\,, \quad \alpha, \beta \in \mathbb{R}\,, \tag{26}$$

and accordingly the element of the Schrödinger group that results from its exponentiation can be parametrized as

$$e^{i\mathcal{L}t} = S(v,w) = e^{(va - \bar{v}a^\dagger)} e^{(\frac{w}{2}a^2 - \frac{\bar{w}}{2}(a^\dagger)^2)}\,, \tag{27}$$

where we have not taken into account bilinears of $a$ and $a^\dagger$ that act diagonally on the Fock states in either of the two expressions involving the Liouvillian, since they just produce a phase that can be included in the normalization. We provide an explicit mapping between the parameters $\alpha, \beta$ and $v, w$ in subsection 3.4.

The Schrödinger group, apart from describing the symmetries of a non-relativistic CFT [51] and thus being relevant for a non-relativistic limit of AdS/CFT [34, 52], finds applications in systems with experimental realizations. In particular, it is used to describe specific processes in molecular dynamics and two-photon processes [35, 53]. Namely, within the study of coherent states, squeezing refers to saturating the uncertainty relation and in quantum optics, to reach such a state, one needs to take into account pairs of photons on top of single photon states. Two-photon processes and the collision of molecules can be modelled with a Hamiltonian expanded in a power series and truncated at quadratic order thus producing an effective Hamiltonian of the form

$$H = \hbar\omega(aa^\dagger + \frac{1}{2}) + f_2(t)(a^\dagger)^2 + f_2^*(t)a^2 + f_1(t)a^\dagger + f_1^*(t)a\,. \tag{28}$$

In the Liouvillian presented above the coefficients $\alpha, \beta$ are time independent, but generally they can be promoted to time-dependent as in this Hamiltonian to capture squeezing that is not necessarily linear in time.

## 3.2 Computation of coherent states

Acting with this general element on a quantum state of a system with Schrödinger symmetry provides a family of generalized coherent states in line with Perelomov [46]. Let us discuss the

construction of those coherent states following [54]. We first consider the Baker-Campbell-Hausdorff (BCH) decomposition of the displacement operator combined with a complex phase factor $\theta$

$$
\begin{aligned}
S(v,w) &= \theta e^{(va - \overline{v}a^\dagger)} e^{\left(\frac{w}{2}a^2 - \frac{\overline{w}}{2}(a^\dagger)^2\right)} \\
&= \theta e^{-\frac{|v|^2}{2}} e^{-\overline{v}a^\dagger} e^{va} e^{-\frac{1}{2}\frac{\overline{w}}{|w|}\tanh(|w|)(a^\dagger)^2} e^{-\ln(\cosh(|w|))(a^\dagger a + \frac{1}{2})} e^{\frac{1}{2}\frac{w}{|w|}\tanh(|w|)a^2} ,
\end{aligned}
\tag{29}
$$

where we have used the standard results for the BCH decomposition of the two exponential terms, see e.g. [35]. One might easily guess that expanding the action of that operator on a general quantum state is impractical. Instead one can obtain a recurrence formula for the coefficients $\phi_k = \langle k | S | 0 \rangle$. However, before that we need to calculate $\phi_0$ which will serve as normalization.

$$
\begin{aligned}
\phi_0 = \langle 0 | S | 0 \rangle &= \theta \frac{e^{-\frac{|v|^2}{2}}}{\cosh(|w|)^{\frac{1}{2}}} \langle 0 | e^{va} e^{-\frac{1}{2}\frac{\overline{w}}{|w|}\tanh|w|(a^\dagger)^2} | 0 \rangle \\
&= \theta \frac{e^{-\frac{|v|^2}{2}}}{\cosh(|w|)^{\frac{1}{2}}} e^{-\frac{1}{2}v^2 \frac{\overline{w}}{|w|}\tanh|w|} .
\end{aligned}
\tag{30}
$$

Subsequently one can compute $\phi_k$ by observing that

$$
\langle k | Sa | 0 \rangle = \langle k | SaS^{-1}S | 0 \rangle = 0 .
\tag{31}
$$

Inserting above the following Bogoliubov transformation (i.e., the result of commuting $a$ with $S^{-1}$)

$$
SaS^{-1} = \cosh|w|a + \frac{\overline{w}}{|w|}\sinh|w|a^\dagger + \overline{v}\cosh|w| + v\frac{\overline{w}}{|w|}\sinh|w| ,
\tag{32}
$$

we obtain the following relation

$$
\sqrt{k+1}\cosh|w|\phi_{k+1} + \sqrt{k}\frac{\overline{w}}{|w|}\sinh|w|\phi_{k-1} + \left(\overline{v}\cosh|w| + v\frac{\overline{w}}{|w|}\sinh|w|\right)\phi_k = 0 .
\tag{33}
$$

This three-term recurrence relation can be solved in terms of Hermite polynomials, thus producing an analytical expression for all $\phi_k$ [54]

$$
\phi_k = \frac{1}{\sqrt{k!}}\left(\frac{1}{2}\frac{\overline{w}}{|w|}\tanh|w|\right)^{\frac{k}{2}} H_k(s)\phi_0 ,
\tag{34}
$$

where

$$
s = -\frac{1}{\sqrt{2}}\left(v\left(\frac{\overline{w}}{|w|}\right)^{\frac{1}{2}}\sqrt{\tanh|w|} + \overline{v}\left(\frac{w}{|w|}\right)^{\frac{1}{2}}\frac{1}{\sqrt{\tanh|w|}}\right) .
\tag{35}
$$

Here, the appearance of orthogonal polynomials is reminiscent of [55], where the authors investigate the relationship of Krylov complexity and orthogonal polynomials. While this hints at a possible connection, the orthogonal polynomials here enter the expression for $\phi_k$ whereas in [55] they play the role of the operator states themselves. It is unclear at this time whether there is an overarching description of this problem in terms of orthogonal polynomials or this is simply a coincidence. Moving past this digression, we have achieved our initial goal of obtaining a distribution over the Fock basis of the Perelomov coherent states for the two-dimensional Schrödinger group in the form

$$
S | 0 \rangle = \sum_{k=0}^{\infty} \phi_k | k \rangle ,
\tag{36}
$$

with the coefficients $\phi_k$ specified above.

### 3.3 Krylov complexity in a natural basis

As we showed previously, the probability distribution over the Fock basis is given by (34). First, let us show that indeed the condition $\sum_k |\phi_k|^2 = 1$ is satisfied. For that we will need to use Mehler's formula

$$\sum_{n=0}^{\infty} \frac{H_n(x) H_n(y)}{n!} \left(\frac{z}{2}\right)^n = (1-z^2)^{-\frac{1}{2}} e^{\frac{2xyz-(x^2+y^2)z^2}{1-z^2}}, \tag{37}$$

which for $x = s$ and $y = \bar{s}$ simplifies to

$$\sum_{n=0}^{\infty} \frac{H_n(s) H_n(\bar{s})}{n!} \left(\frac{z}{2}\right)^n = (1-z^2)^{-\frac{1}{2}} e^{\frac{2|s|^2 z-(s^2+\bar{s}^2)z^2}{1-z^2}}. \tag{38}$$

The sum expression $|\phi_n|^2$ in terms of (34), using that $\overline{H}_n(z) = H_n(\bar{z})$, simply becomes

$$\sum_{k=0}^{\infty} \frac{H_k(s) H_k(\bar{s})}{k!} \left(\frac{\tanh |w|}{2}\right)^k |\phi_0|^2 = \cosh |w| e^{\frac{2|s|^2 \tanh |w| - (s^2+\bar{s}^2)\tanh^2 |w|}{1-\tanh^2 |w|}} |\phi_0|^2 = 1, \tag{39}$$

where we used the definition of $s$ from (35) to show the last step. In computing Krylov complexity we consider rewriting the definition as follows

$$\begin{aligned}
K_{\mathcal{O}} &= \sum_{k=0}^{\infty} k |\phi_k|^2 \\
&= |\phi_0|^2 z \partial_z \sum_{k=0}^{\infty} \left(\frac{z}{2}\right)^k H_k(s) \overline{H}_k(s) - |\phi_0|^2 z \sum_{k=0}^{\infty} \left(\frac{z}{2}\right)^k \partial_z \left[H_k(s) \overline{H}_k(s)\right] \\
&= |\phi_0|^2 z \partial_z |\phi_0|^{-2} - |\phi_0|^2 y \partial_y |\phi_0|^{-2} \\
&= |v|^2 + \sinh^2 |w|,
\end{aligned} \tag{40}$$

where $z = \tanh |w|$ and $y = (v/|v|)^2$ and in the third equality we used the identity

$$z \partial_z s = y \partial_y s. \tag{41}$$

In fact, using induction one can easily arive at the conclusion

$$K_{(n)} = \sum_{k=0}^{\infty} k^n |\phi_k|^2 = |\phi_0|^2 \left(\frac{\sinh(2|w|)}{2} \partial_{|w|}\right)^n |\phi_0|^{-2} \Big|_s, \tag{42}$$

where $n$ is any integer and $|_s$ indicates that when taking the derivative with respect to $|w|$, $s$ is kept fixed.

The variance $\sigma^2 = \langle k^2 \rangle - \langle k \rangle^2$ can be computed using the identity in (42) and reads

$$\sigma^2 = |v| \cosh 2|w| + \sinh |w| \cosh |w| \left(\sinh 2|w| - \frac{\bar{v}^2 w + v^2 \overline{w}}{|w|}\right). \tag{43}$$

To get some more feeling for expression (40), it is instructive to reproduce the limit in which either the Heisenberg-Weyl part or the $SL_2(\mathbb{R})$ part are turned off by hand. If we take $v = -\bar{v} = i\alpha t$ and $w = \overline{w} = 0$ we find

$$K_{\mathcal{O}} = \alpha^2 t^2, \tag{44}$$

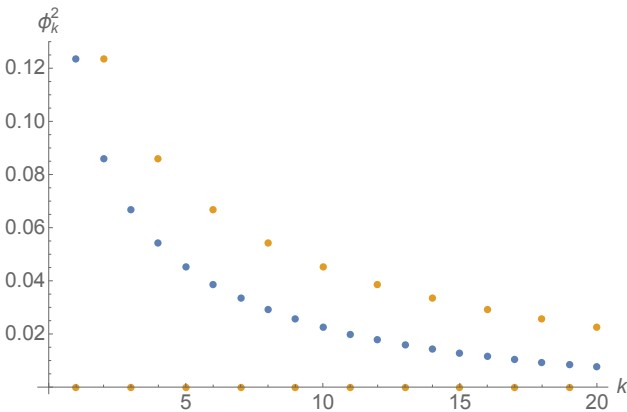

Figure 1: In this figure we compare the probabilities for the Schrödinger group (yellow dots) and $SL_2(\mathbb{R})$ (blue dots) as a function of their index $k$. Notice that for odd $k$ the probabilities for the Schrödinger group vanish, while for even $k$ they are pairwise equal to $SL_2(\mathbb{R})$, but shifted.

where we assumed $\alpha$ to be real and we indeed reproduce the correct Krylov complexity from (13). In the opposite case we consider $v = \bar{v} = 0$ and $w = -\bar{w} = i\beta t$, where $\beta$ is real, and we find

$$K_{\mathcal{O}} = \sinh^2 \beta t, \tag{45}$$

which indeed reproduces (18) (with $h = 1/4$) up to a factor of 2. This can be attributed to the fact that the chain picture that we obtain for the Schrödinger group has double the number of sites compared to $SL_2(\mathbb{R})$. Setting $v = 0$ has the effect of the probability vanishing on odd sites, which however are still counted by the complexity. On the contrary the way that the Krylov basis is built for $SL_2(\mathbb{R})$ (using the harmonic oscillator realization of the algebra presented previously) is such that only the even sites are taken into account to begin with and this subtle difference leads to this factor of two discrepancy. This effect is captured graphically in figure 1.

### 3.4 Interpreting complexity

The displacement operator $S(v, w)$ introduced in (29) leads to Perelomov coherent states for the two-dimensional Schödinger group. In order to interpret the result for complexity we need to relate the displacement operator $S(v, w)$ to the Liouvillian

$$\mathcal{L} = \alpha(a^\dagger + a) + \frac{\beta}{2}\left((a^\dagger)^2 + a^2\right), \quad \alpha, \beta \in \mathbb{R}, \tag{46}$$

such that complex parameters $v$ and $w$ inherit the appropriate time dependence. To relate these quantities we will use the identity

$$e^{i[\alpha(a^\dagger + a) + \frac{\beta}{2}((a^\dagger)^2 + a^2)]t} = \theta\, e^{(va - \bar{v}a^\dagger)} e^{(\frac{w}{2}a^2 - \frac{\bar{w}}{2}(a^\dagger)^2)}, \tag{47}$$

$$v = \frac{\alpha}{\beta}(1 - \cosh\beta t) + i\frac{\alpha}{\beta}\sinh\beta t, \quad w = i\beta t, \tag{48}$$

and we pick up a phase factor $\theta = \exp\left[i\frac{\alpha^2}{\beta^2}(\sinh\beta t - \beta t)\right]$. To obtain the above identity we circumvented using the BCH formula by adopting an explicit matrix representation [35] that

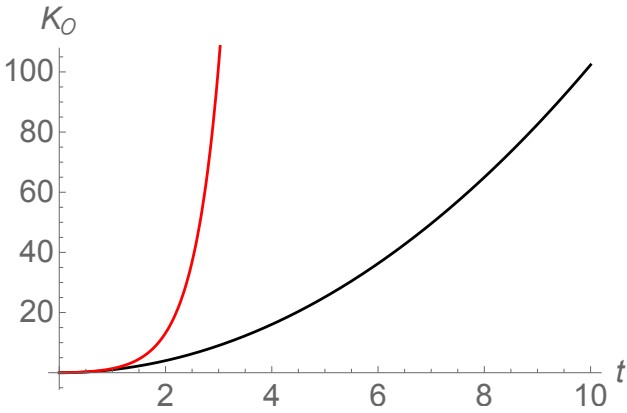

Figure 2: This figure contains the plots of complexity as a function of time for the cases where $\alpha = 0.01, \beta = 1$ (red curve) and $\alpha = 1, \beta = 0.01$ (black curve). We observe that the red curve clearly exhibits the exponential behavior characteristic of $SL_2(\mathbb{R})$ whereas the black curve is closer to the quadratic behavior of $HW$.

allows us to perform ordinary matrix exponentiation:

$$\eta\left(a^\dagger a + \frac{1}{2}\right) + \delta + R(a^\dagger)^2 + La^2 + ra^\dagger + la \quad \mapsto \quad \begin{pmatrix} 0 & 0 & 0 & 0 \\ r & \eta & 2R & 0 \\ -l & -2L & -\eta & 0 \\ -2\delta & -l & -r & 0 \end{pmatrix}, \qquad (49)$$

where $\eta$, $\delta$, $R$, $L$, $r$, $l$ are complex numbers.

Now that we know $v(t)$ and $w(t)$ we can interpret complexity (40) as a function $t$:

$$K_{\mathcal{O}} = \alpha^2 t^2 + \sinh^2 \beta t + \alpha^2 \left[ \frac{4\cosh \beta t \sinh^2 \frac{\beta t}{2}}{\beta^2} - t^2 \right], \qquad (50)$$

where the first term reproduces the $HW$ complexity ($\beta = 0$) and the second term reproduces the $SL_2(\mathbb{R})$ complexity ($\alpha = 0$), for a comparison see Section 2. The third term in square brackets, which we dub the interaction term, vanishes in either cases and we interpret it as arising from the fact that elements of either sub-algebras do not commute. When expanding the hyperbolic functions in the interaction term we get an infinite series in even power that cancels the single $t^2$, showing that the interaction term is always positive. This means that the Schrödinger complexity is more than the sum of the separate complexities of $HW$ and $SL_2(\mathbb{R})$. While for non-zero $\alpha, \beta$ the interaction term is always present, the relative size of these parameters determines the character of the system. In other words if $\frac{\alpha}{\beta}$ is small then the system behaves closer to pure $SL_2(\mathbb{R})$ and conversely when $\frac{\alpha}{\beta}$ is large then the behavior resembles pure $HW$, as seen in figure 2.

Let us consider early and late time limits to explore the effects of the interaction term. For early time we find

$$K_{\mathcal{O}} = (\alpha^2 + \beta^2)t^2 + \mathcal{O}(t^3), \qquad (51)$$

where the $\beta^2$ comes from the $\sinh^2$ and the interaction term does not contribute. The effects of the interaction term becomes apparent at late times

$$K_{\mathcal{O}} = \left(\frac{1}{4} + \frac{1}{2}\frac{\alpha^2}{\beta^2}\right)e^{\beta t} = e^{\beta(t - t_s)}, \qquad (52)$$

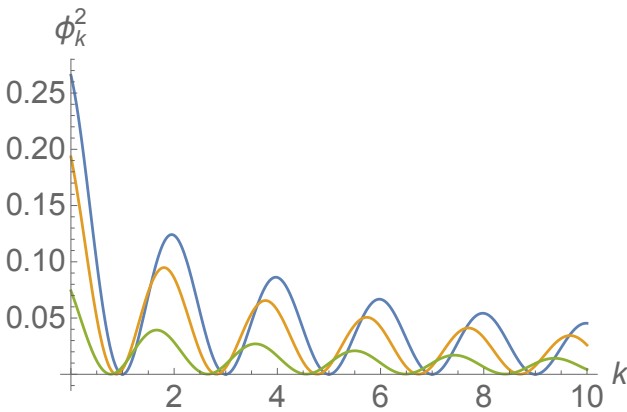

Figure 3: Probabilities for the Schrödinger group as functions of their index k (made into a continuous variable for illustration purposes) for $\alpha = 0$ (blue curve), $\alpha = 0.25$ (yellow curve) and $\alpha = 0.5$ (green curve). We notice that the magnitude of the probability drops for larger values of $\alpha$. Since the probability functions are normalized. This illustrates the faster spreading that occurs due to the interaction term in (54).

which implies that the Lyapunov exponent remains unaltered, but the scrambling time $t_s$, which can be a probe for operator growth, does receive a correction:

$$t_s = \frac{1}{\beta} \log \frac{4\beta^2}{\beta^2 + 2\alpha^2} \,. \tag{53}$$

For $\alpha > 0$, the scrambling time decreases as compared to the $SL_2(\mathbb{R})$ case and for $\alpha > \beta\sqrt{3/2}$ the scrambling time becomes negative, as can also happen for the pure $SL_2(\mathbb{R})$ case for different representations, i.e., values of $h$ in (18). This is to be expected as it signals that one needs more information than just the leading order term for such cases. We furthermore point out that the first and second derivative of the complexity with respect to time are positive for $t \geq 0$.

The variance in (43) combined with $v(t)$ and $w(t)$ and normalized with $K_{\mathcal{O}}^2$ yields

$$\sigma^2 = \left\{ \alpha^2 t^2 + \frac{1}{2}\sinh^2 2\beta t + \frac{\alpha^2}{2\beta^2}\left[ 4(\cosh\beta t + \cosh 3\beta t)\sinh^2\frac{\beta t}{2} - 2\beta^2 t^2 \right.\right.$$

$$\left.\left. + 32\cosh\frac{\beta t}{2}\cosh^{3/2}\beta t \sinh^4\frac{\beta t}{2}\right]\right\} \bigg/ \left[ \sinh^2\beta t + 4\frac{\alpha^2}{\beta^2}\cosh\beta t \sinh^2\frac{\beta t}{2}\right]^2, \tag{54}$$

where the first two terms correspond to the variance of purely $HW$ and $SL_2(\mathbb{R})$ respectively. The term in the numerator within brackets is positive for $t > 0$. In order to appreciate the effect of the interaction term it is useful to investigate the plots of the probabilities $\phi_k^2$ which appear in figure 3.

It is also worth examining in more detail the form of $|\phi_0|^2$, which is the autocorrelator (survival amplitude) of the system

$$|(\mathcal{O}(0)|\mathcal{O}(t))|^2 = |\phi_0(t)|^2 = \frac{e^{-\frac{\alpha^2(e^{2\beta t}-1)^2}{8\beta^2}+2\beta t}}{\cosh(2\beta t)}, \tag{55}$$

where we have restored the time dependence using (47). We remind the reader that the information contained in the autocorrelation function is equivalent to the Lanczos coefficients, or in our case the hopping coefficients. We observe that the leading contribution to the decay

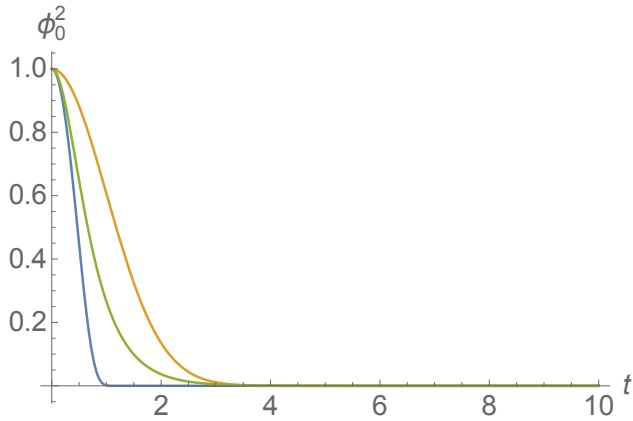

Figure 4: Comparison for the autocorrelator functions for $\alpha = 1, \beta = 0$ (yellow curve), $\alpha = 0, \beta = 1$ (green curve), $\alpha = 1, \beta = 1$ (blue curve). We observe that the autocorrelator for the Schrödinger group decays faster than both the $HW$ and $SL_2(\mathbb{R})$ groups.

of the autocorrelation function is of the form $e^{-\alpha^2 e^{2\beta t}}$, that is doubly exponential. This is unlike any of the semisimple groups that were studied so far, although it corroborates the enhanced spreading of the wavefunction as argued previously. We illustrate this result by comparing to the results for pure $HW$ and $SL_2(\mathbb{R})$ in figure 4.

## 4 Discussion

In this paper we proposed an approach to computing Krylov complexity for the two-dimensional Schrödinger group in what we dub to be a *natural* orthonormal basis, which does not involve the usual tridiagonal Liouvillian, but a pentadiagonal one instead. We argue that regardless the same Krylov subspace is probed. The naturalness of the basis arises from the fact that the two-dimensional Schrödinger group is generated by a non-semisimple algebra that can be written as the semi-direct sum of the Heisenberg-Weyl algebra and $SL_2(\mathbb{R})$, which can both be naturally expressed in the Heisenberg-Weyl basis. We advocate that this approach might provide insights to other non-semisimple algebras for which the usual Krylov basis is technically non-attainable.

We find that the Krylov complexity in its natural basis is greater than the sum of the Krylov complexity of the separate sub-algebras and the same holds for its variance. At late time we recover the same Lyapunov exponent as for $SL_2(\mathbb{R})$, but we do find a naive scrambling time of smaller value. This is an intriguing aspect of our results as it creates the following picture. At late times the dynamics of the system are dominated by the $SL_2(\mathbb{R})$ degrees of freedom, which is why we observe the exponential growth of Krylov complexity with the same Lyapunov exponent. However, the spreading of the wave function, or in other words the distribution of the state over the Krylov subspace happens at a faster rate as shown by our variance calculations and the smaller scrambling time. While a rigorous interpretation of this phenomenon is outside of our reach for the moment, this highlights the importance of probing multiple aspects of the probability distribution rather than just the average. We believe this behavior makes sense from an intuitive point of view: due to its semi-direct sum structure, elements in different sub-algebras have non-trivial commutation relations and as such there should be an increase of the total complexity compared to the naive sum of the sub-algebras.

Our complexity grows convexly, which contrasts the numerical findings of [36], who considered Krylov complexity in an approximate tridiagonal basis. They furthermore explicitly find that the complexity of the two-dimensional Schrödinger algebra is *less* than its separate sub-algebras. It would be interesting to understand this discrepancy, especially since in the end both methods probe the same Krylov subspace.

Doing computations in the natural basis, from the perspective of the harmonic oscillators basis $a^\dagger$ and $a$, amounts to taking into account generators $(a^\dagger)^2$ and $a^2$ in the Liouvillian. One might wonder: why stop at quadratic order? It turns out that higher orders, say, $a^{\dagger 3}$ and $a^3$ do not close the algebra. Commuting these generators yields fourth order generators in terms of the oscillator basis and one is led to conclude that the algebra does not close for a finite number of generators.

The autocorrelator computed using the Krylov basis for $SL_2(\mathbb{R})$ can be reproduced using a thermofield double setup in the context of holography. The Schrödinger algebra is non-Lorentzian and its holographic manifestation is less well understood [34,56]. Perhaps the here presented results, the autocorrelator that exhibits doubly-exponential scrambling specifically, can provide a bench mark for further developing Schrödinger holography.

Finally, it is important to ask what our results imply for the bigger picture and how do they fit into our understanding of Krylov complexity and its relation to other complexity measures. Here we have established that one can obtain analytic results for systems with symmetry that go beyond the case of semisimple Lie algebras. The key ingredient in this endeavour is the use of generalized coherent states, which appear to play a prevalent role not only when it comes to Krylov complexity, but also in approaches to complexity that rely on geometry as is the case in [57–62] for example. This is due to the natural organization of coherent states in metric spaces as explained in detail in [46]. It would be interesting to think how our results fit into that framework as the Schrödinger group provides an incremental step in tackling more complicated problems like the Virasoro group which, being centrally extended, also falls into the category of non-semisimple groups. Due to its relevance in AdS/CFT this group has been studied extensively and there are some results with regard to complexity that take advantage of the associated geometry and are possibly implicitly related to certain classes of coherent states [59]. Recently there have also been advances in understanding Krylov complexity from a holographic perspective [37,47,63], although there are still many open questions that one would hope to address. Thus, we believe that our work is not only important for a non-relativistic limit of AdS/CFT, but also provides the groundwork for more general considerations.

# Acknowledgments

It is a pleasure to acknowledge the scientific program "Reconstructing the Gravitational Hologram with Quantum Information", which took place at the INFN Galileo Galilei Institute (GGI) for Theoretical Physics, National Centre for Advanced Studies in the summer of 2022, in Florence, where this work was initiated. We furthermore wish to thank Paweł Caputa for insightful discussions and support and Jose Barbon, Jan Boruch, Sinong Liu and Javier Magan for helpful comments.

**Funding information** DP is supported by "Polish Returns 2019" grant of the National Agency for Academic Exchange (NAWA) PPN/PPO/2019/1/00010/U/0001 and Sonata Bis 9 2019/34/E/ST2/00123 grant from the National Science Center, NCN. WS is supported by the Icelandic Research Fund via the Grant of Excellence titled "Quantum Fields and Quantum Geometry" and by the University of Iceland Research Fund. WS furthermore thanks Paweł and Dimitris for their unrivalled hospitality in Warsaw.

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
