# Peer review of "Krylov complexity in a natural basis for the Schrödinger algebra"

_SciPost Physics, doi:SciPost Phys. Core 7, 037 (2024)_

## Round 2 · Referee Report · Anonymous (Referee 1) · 2023-8-22

Report
This work suggests using a different construction for the Krylov basis for a specific case where the Liouvillian can be described as a general element in the 2-dimensional Schrödinger (Lie) algebra. The authors show that the resulting "Krylov complexity" is greater than the sum of the Krylov complexity of the two sub-algebras which make up the Schrödinger algebra via a semi-direct sum. The Liouvillian the authors propose to use is penta-diagonal instead of tri-diagonal as a result of using an algebra-adapted orthonormal basis as the Krylov basis. The algebraic structure makes the solution for the wavefunctions $\phi_k(t)$ possible to perform analytically, using coherent states, and from them the associated "K-complexity" is computed.
I find the work interesting, but I have several comments which I believe the authors should address.
General comments: 1. The original notion of K-complexity as defined and studied in [1, 39] is general for any hermitian Hamiltonian and operator, and it does not seem to need a re-definition. In particular in [39, 38] it was shown to satisfy the general behaviour expected from complexity of operators at all time scales, including saturation at time scales exponential in the number of DoF. 2. It is not clear to me how this approach could be generalized to other systems where the above-mentioned algebra cannot be used as a guideline. This is also related to the first point: could it be shown that such a definition follows the general behaviour of complexity, or otherwise, and what properties will it have? 3. Did the authors try probing the usual Krylov basis and complexity for the system they study? Even numerically? For example via the autocorrelation function (55).
It would be appropriate if the authors mention these points and discuss them further in their work.
More detailed comments: 1. Below equation (4) the authors say that the tri-diagonal structure of the Liouvillian is a result of the properties of the inner-product. In fact, as the authors can check, to have zeros on the diagonal the operator should also be hermitian. 2. I believe the authors should cite the work "Building Krylov complexity from circuit complexity" by Lv, Zhang and Zhou in relation to the use of coherent states to study Krylov complexity as well as in the discussion of the relation of K-complexity to other notions of complexity. 3. It would be good if the authors provide more details on the "Krylov" basis they construct and not only on the wavefunctions. 4. Unlike the statement the authors made about reference [47] by Rabinovici et al, [47] has indeed established a rigorous bulk-boundary holographic duality between K-complexity and a geometric bulk quantity, in the particular setup of 2-dimensional JT gravity.

Author: Watse Sybesma on 2023-10-09 [id 4033]
(in reply to Report 1 on 2023-08-22)We thank the reviewer for their insightful comments, which we address below individually (we've added a modified draft to reflect changes accordingly):
Comment by referee:
"The original notion of K-complexity as defined and studied in [1, 39] is general for any hermitian Hamiltonian and operator, and it does not seem to need a re-definition. In particular in [39, 38] it was shown to satisfy the general behaviour expected from complexity of operators at all time scales, including saturation at time scales exponential in the number of DoF. "
Response:
The aim of our work is not to redefine Krylov complexity as a general concept. Indeed it has a perfectly good definition which can, in a number of cases, capture the essential features of operator spreading at all time scales. Our aim is to provide an alternative yet closely related measure which can still be analytically computed in cases when the implementation of the Lanczos algorithm is not tractable. Given that this measure is still probing the Krylov subspace defined by the Liouvillian, we believe it is accurate to use the term Krylov complexity with the additional remark that it is computed in what we dub as the “natural basis”, as we explain both in the introduction and in the discussion section.
Comment by referee:
"It is not clear to me how this approach could be generalized to other systems where the above-mentioned algebra cannot be used as a guideline. This is also related to the first point: could it be shown that such a definition follows the general behaviour of complexity, or otherwise, and what properties will it have? "
Response:
Generally more complicated symmetries are not always expressible in terms of the raising and lowering operators which we are indeed using as a guideline in our work. However, the approach itself which consists of constructing a Liouvillian (or Hamiltonian for state complexity) out of the algebra generators and considering its action on a given state can be implemented for arbitrary Lie groups. Of course one might expect that this will lead to more complicated constructions, for example more elaborate dynamics on the Krylov chain or more involved forms of coherent states, but we still believe that it can serve as a useful tool that provides us with a handle for analytic calculations in cases which this is otherwise not possible. In terms of the properties of the complexity we are computing, we have shown that in certain limits and regimes its behaviour matches what we expect from Krylov complexity. For a rigorous understanding of the relationship between the actual Krylov basis and the “natural basis”, it is necessary to have access to the formal so it is not possible to check directly, but the checks that we have performed hint that the time dependence of complexity should be the same in both cases up to a multiplicative constant and so the essential features of operator growth should be the same. However, in general our approach would apply to any non-semi-simple sum of Lie algebra’s that can be expressed in a common basis.
Comment by referee:
"Did the authors try probing the usual Krylov basis and complexity for the system they study? Even numerically? For example via the autocorrelation function (55)"
Response:
As we mention in the paper this has been the subject of reference [36] in which the authors provide an approximation scheme which numerically computes the Krylov basis and complexity. We compare our results to theirs in the discussion section. The autocorrelation function (55) can be used to compute the moments and Lanczos coefficients, which however appear to be problematic since they become complex. This is not surprising since the autocorrelation function is not Fourier transformable because it is doubly exponential and so obtaining the power spectrum which is directly related to the Lanczos coefficients would require a much more careful treatment. Potentially a more refined numerical approach could resolve the issue, but this is beyond the scope of our work.
Comment by referee: "Below equation (4) the authors say that the tri-diagonal structure of the Liouvillian is a result of the properties of the inner-product. In fact, as the authors can check, to have zeros on the diagonal the operator should also be hermitian."
Response:
We corrected the statement as suggested.
Comment by referee:
"I believe the authors should cite the work "Building Krylov complexity from circuit complexity" by Lv, Zhang and Zhou in relation to the use of coherent states to study Krylov complexity as well as in the discussion of the relation of K-complexity to other notions of complexity."
Response:
We added the reference as suggested.
Comment by referee:
"It would be good if the authors provide more details on the "Krylov" basis they construct and not only on the wavefunctions."
Response:
The basis that we use is simply the Fock basis on which the Liouvillian acts naturally. In other words we exploit the fact that the algebra of raising and lowering operators is associated with an automatically orthonormal number basis which we can use as a probe for the Krylov subspace. Part of the appeal of our approach is the fact that we do not have to construct the basis, but it is rather automatically provided to us by the algebra.
Comment by referee:
"Unlike the statement the authors made about reference [47] by Rabinovici et al, [47] has indeed established a rigorous bulk-boundary holographic duality between K-complexity and a geometric bulk quantity, in the particular setup of 2-dimensional JT gravity."
Response:
It is true that Rabinovici et al establish a rigorous correspondence between the length eigenstates in the bulk and the Krylov basis in the triple scaling limit of SYK. However, this does not constitute a general entry into the holographic dictionary, although one would hope that I can be generalized to setups in higher dimensions. For example, already for asymptotically AdS2 models beyond JT it would be worthwhile to check this entry into the holographic dictionary and establish this duality even further. Regardless, we have altered the statement in the paper to eliminate any ambiguity.
Attachment:
Krylov_complexity_in_a_natural_algebra_basis.pdf

---

## Round 3 · Referee Report · Anonymous (Referee 1) · 2024-2-21

Report

I thank the authors for the detailed clarifications. Before recommending the manuscript for publication in SciPost, I would suggest that the authors add these clarifications to the actual manuscript. This would make their work much more valuable to future readers.

---

## Round 4 · Referee Report · Anonymous (Referee 2) · 2024-5-22

Strengths

gives new explicit calculations of Krylov complexities by exploiting a symmetry-based approach.

Weaknesses

  1. The scope of the results limited
  2. Only limited physical interpretation given by authors

Report

Krylov complexity is a relatively recent measure of quantum complexity, whose basic phenomenology is being actively investigated. The present article adds to this field of study by extending the formalism to theories with Schrödinger group symmetry, and argue that their method more generally lets them compute Krylov complexity for symmetry groups that posses a semi-direct sum structure.

The use of the so-called “natural basis” in which the Liouvillian is penta-diagronal is a departure from the usual Krylov algorthithm, although the authors argue that the same Krylov space is still probed.

I find the results interesting and the responses and changes made in response to another referee’s comments to have improved the paper sufficiently to merit publication. Due to the somewhat limited scope of the analysis I would, however, recommend publication in SciPost Core.

Recommendation

Accept in alternative Journal (see Report)

---

## Round 4 · Author Response

We fixed typos, amongst which eq. 8.

---

## Editorial Decision

published